# Communication Experiences in Primary Healthcare with Refugees and Asylum Seekers: A Literature Review and Narrative Synthesis

**DOI:** 10.3390/ijerph18041469

**Published:** 2021-02-04

**Authors:** Pinika Patel, Sarah Bernays, Hankiz Dolan, Danielle Marie Muscat, Lyndal Trevena

**Affiliations:** 1Sydney School of Public Health, Faculty of Medicine and Health, The University of Sydney, Sydney, NSW 2006, Australia; sarah.bernays@sydney.edu.au (S.B.); hankiz.dolan@sydney.edu.au (H.D.); lyndal.trevena@sydney.edu.au (L.T.); 2Ask Share Know: Rapid Evidence for General Practice Decisions (ASK-GP), Centre for Research Excellence, Sydney School of Public Health, Faculty of Medicine and Health, The University of Sydney, Sydney, NSW 2006, Australia; danielle.muscat@sydney.edu.au; 3Department of Global Health and Development, Faculty of Public Health and Policy, London School of Hygiene and Tropical Medicine, London WCIE 7HT, UK; 4Sydney Health Literacy Lab, Sydney School of Public Health, Faculty of Medicine and Health, The University of Sydney, Sydney, NSW 2006, Australia

**Keywords:** refugees, asylum seekers, primary healthcare, general practice, communication, patient-centered care, patient engagement

## Abstract

Refugee and asylum seeker population numbers are rising in Western countries. Understanding the communication experiences, within healthcare encounters, for this population is important for providing better care and health outcomes. This review summarizes the literature on health consultation communication experiences of refugees and asylum seekers living in Western countries. Seven electronic databases were searched from inception to 31 March 2019. Studies were included if they aimed to improve, assess or report on communication/interaction in the primary health care consultation setting with refugees or asylum seekers, and were conducted in Western countries. A narrative synthesis of the literature was undertaken. Thematic analysis of the 21 included articles, showed that refugees and asylum seekers experience a range of communication challenges and obstacles in primary care consultations. This included practical and relational challenges of organizing and using informal and formal interpreters and cultural understanding of illness and healthcare. Non-verbal and compassionate care aspects of communication emerged as an important factor in helping improve comfort and trust between healthcare providers (HCP) and refugees and asylum seekers during a healthcare encounter. Improvements at the systems level are needed to provide better access to professional interpreters, but also support compassionate and humanistic care by creating time for HCPs to build relationships and trust with patients.

## 1. Introduction

There are currently 70.8 million forcibly displaced people worldwide, with approximately 37,000 people displaced every day [1]. This includes refugees “who have fled war, violence, conflict or persecution, have crossed an international border and been granted protection/safety” and asylum seekers “who have sought international protection and whose claims for refugee status have not yet been determined” [2]. Many of the refugees and asylum seekers arrive in Western resettlement countries with complex psychological and physiological health needs. They face challenges accessing and utilizing healthcare due to numerous factors, such as unfamiliarity with the healthcare system, language and cultural barriers, cost and other social circumstances [3,4,5,6].

Primary healthcare services are usually refugee and asylum seekers’ first point of care in the resettlement countries [3,5,7,8]. Such services often face challenges in not only training healthcare providers (HCP) in effectively responding to the healthcare needs of the refugee and asylum seeker patients but also in identifying issues with patient’s immigration status and access to healthcare [9]. Apart from these broader system-level challenges, another key area where challenges arise is the healthcare encounter between refugee and asylum seekers and HCP [9]. Communication plays a key role in the healthcare encounter between refugee and asylum seekers and healthcare providers and is an essential starting point for patient satisfaction and positive health outcomes [10].

Experiences within the healthcare encounter, in particular the interpersonal relationships, are fundamental to good healthcare provision [11,12]. Clinician–patient relationships and patient health outcomes rely on effective communication between the clinician and patient [10]. When considering people from culturally and linguistically diverse backgrounds, communication has been identified as the starting point for building up confidence between the healthcare provider and patient [13]. Evidence has shown that patient satisfaction is strongly associated with communication behaviors during the clinician–patient interaction [14,15,16].

This aim of this review is to summarize the literature on the communication experiences of refugee, asylum seekers and healthcare providers during primary healthcare consultations in Western countries (defined by UN regional grouping) in order to inform recommendations for practice [17].

## 2. Methods

This review summarizing current research on communication experiences is guided by a systematic literature searching methodology [18] with narrative data synthesis and analysis techniques [19].

### 2.1. Search Strategy

Seven electronic databases were systematically searched from inception to 31 March 2019: OVID Medline, EMBASE, CINAHL, Web of Science, Scopus, Global Health and Informit.

Search terms for primary healthcare, refugees and asylum seekers and communication were combined to develop the search strategy (Appendix A). No date limits were applied, but studies were limited to those with titles and abstracts in English. Further hand-searches were conducted based on included studies’ reference lists and citations (in Google Scholar).

After the removal of duplicates using Endnote X8 software (Clarivate Analytics, Philadelphia, PA, USA), the remaining references were imported to the Rayyan online tool [20] for screening and data extraction. The titles and abstracts were screened by two researchers, excluding articles that did not clearly meet the pre-defined inclusion criteria. The full texts of the remaining articles were obtained and assessed by two independent researchers, according to prespecified study selection criteria (detailed below). Any disagreements were resolved via discussion. Where full texts were not in English, native speakers completed the screening process. Full texts of studies which met the pre-specified study selection criteria were translated into English using Google Translate and proofread by native speakers prior to data extraction.

Studies were excluded if the full-text could not be obtained either through institutional access or from requests sent to authors through Research Gate.

### 2.2. Selection Criteria

#### 2.2.1. Population

Studies were included if participants were refugees and asylum seekers living in Western countries (defined as countries that are members of UN classification of Western European and Other States Group (WEOG)) [17]. Studies were limited to Western countries because of the authors’ interests in developing recommendations for practice applicable to primary healthcare systems in this context.

The literature that presented a mixed population broader than refugees and asylum seekers was excluded, as were studies which referred to “migrants” or “immigrants” but had no information on the migration pathway. Studies regarding “Undocumented migrants,” defined as anyone residing in any given country without legal documentation, were also excluded as this population is known to have unique characteristics that would not necessarily be typical of refugees and asylum seekers [21].

#### 2.2.2. Study Design

Empirical quantitative studies and qualitative studies, case reports, mixed-method studies, reports and opinion articles were included in the review.

Studies designed to improve, assess or report on communication in the primary healthcare consultation setting were included. The definition of the “primary healthcare provider team” is diverse; hence this review was limited to the literature involving the following clinical healthcare providers (HCP): general practitioners (GPs), nurses and midwives. The literature including mental health professionals was also excluded as this clinical area has specific characteristics that shape the communication context.

Studies were excluded if the setting was not within a healthcare encounter or if it was related to accessing healthcare.

### 2.3. Data Extraction and Quality Assessment

Study characteristics were extracted by one author using a data extraction proforma. Characteristics included country of origin, aims, participants, setting, study design, methodology, results and recommendations/applications.

The quality of the included literature was assessed using the respective Joanna Briggs Institute critical appraisal checklists for qualitative research (10-item checklist), text and opinion papers (6-item checklist), studies reporting prevalence data (9-item checklist) and case reports (8-item checklist) [22].

### 2.4. Data Analysis and Synthesis

Qualitative and quantitative methodologies are varied in nature; therefore, a narrative synthesis of the literature was undertaken and involved using inductive thematic analysis in which dominant and recurrent themes were identified. The narrative synthesis described by Popay et al. [19] was used in guiding the process. The analysis involved generating codes from the literature to identify key ideas and then identifying the themes by grouping the codes with similar ideas together. The relevant codes which aligned with the initial research question were all incorporated into themes. We also used grouping and tabulation methods for preliminary synthesis of the study characteristics.

## 3. Results

The systematic database searches identified 4692 articles. Twelve further articles were identified through hand-searching of reference lists and citations. After the removal of duplicates, 2676 articles remained. A further 2588 articles were removed after screening of the title and abstracts. Full texts of the remaining 88 articles were obtained and assessed against inclusion criteria. Full texts could not be obtained for five of the articles. After reviewing the 83 available full texts articles, 21 articles were included in the narrative synthesis (Figure 1). This included sixteen qualitative studies, two opinion articles, two quantitative studies and one case report.

The included articles were conducted in nine countries; six articles were from Australia, four from the Netherlands, three from the United States, two from Ireland, two from Scotland and one each from Norway, Denmark, Sweden, and Canada. All articles were published between 1999 and 2018 (Table 1).

The studies represented the experiences of a total of 357 patient participants and 231 healthcare providers. Of the eighteen qualitative and quantitative studies, nine looked at patient experiences of communication in healthcare setting, six looked at HCP experiences and three looked at HCP and patient perspectives. Both the quantitative studies looked at the HCP experiences. Patient participants were described as “refugees” in seventeen articles, “asylum seekers” in two articles and “refugees and asylum seekers” in two articles.

The Joanna Briggs Institute critical appraisal checklists scores for qualitative studies ranged from 6 to 9 (out of 10), the case report was 5 (out of 8), the studies reporting prevalence data ranged from 7 to 8 (out of 9) and the opinion articles were 6 (out of 6). All of the studies were deemed to be of high quality, so were all included in the literature review (Table 1). Table 2 identifies the study aims, objectives and outcome measures of included studies.

Three themes were identified from the included literature from both the patient and healthcare provider perspectives: (a) linguistic barriers, (b) clinician cues and (c) cultural understanding. The included quantitative studies focused only on linguistic barriers whilst the other study types had elements of all three themes.

### 3.1. Linguistic Barriers

Linguistic barriers were identified through the qualitative and quantitative studies, opinion articles and the case report. This theme emphasized the challenges stemming from the discordance of language between the patient and HCP as well as the difficulties of organizing and using both professional and informal (family and friends) interpreters.

#### 3.1.1. Qualitative Studies

Across studies, accessing appropriate interpreters in a timely manner was one of the prominent challenges highlighted by HCPs. In particular, those with limited experience working with migrants were not always aware of available interpreting services (e.g., telephone services) and the time required to organize an interpreter before the consultation with the patient [32,34,35,40].


*“The times that I have needed it they have been–appointments have been booked well in advance. How do you book an interpreter when someone rings up at lunchtime and sees you two hours later for something that is minor or insignificant?”*
—HCP [34]

Generally HCPs felt that professional interpreters were more experienced with medical terminology and, therefore, provided better outcomes [23,32,33,41]. In the absence of professional interpreters, they used family or friends as interpreters for clinical consultations but expressed their concerns about safety, confidentiality and accuracy of translation [23,25,32,33,36].


*“Sometimes it is okay, but in the majority of the cases it is better with the authorized interpreters since they are more familiar with the medical terminology. So it is always a poorer consultation. It is typically the family being used and I feel they shouldn’t be there at all”*
—HCP [33]

Patients often reported that they were not confident using interpreters due to fear that their problems would not remain confidential and would become gossip. This caused them to be less open with their HCP [31,35]. HCPs also reported that often patients would choose to have a consultation without an interpreter due to the interpreter being known in the community [38].


*“Sometimes you will see a client who does not want to work with an interpreter, especially in small communities there are limited numbers of interpreters from that community. The client may know the interpreter or know people who know the interpreter and they will worry about confidentiality. That causes a lot of embarrassment for women…”*
—HCP [38]

Miscommunication with both professional and informal interpreters (e.g., family and friends) was also seen as an issue by patients in several studies as they sometimes felt that the translations were not correct or the language the interpreter was using was slightly different to their own. HCP experiences in some studies also showed that they were apprehensive about the translation as patients often spoke for an extended period but the responses received through the interpreter were relatively short [24,25,31].


*“When you get a translator and the translator doesn’t really get you the translation in details. Some of them just talk and talk and then when it comes to the translator, he can’t put the words the [right] way...”*
*—patient* [24]

In the absence of interpreters and with limited language skills, patients expressed that they sometimes did not understand the information and explanations that the HCP had given. However, they did not often express this and hence left with unresolved questions and, in some instances, incorrect diagnoses [31,35].


*“Inevitably there were misunderstandings during her GP consultations and, on one occasion, her son who had diarrhoea was prescribed medication for constipation...”*
*—patient* [35]

#### 3.1.2. Quantitative Studies

A survey of 38 HCPs in the United States showed that HCP’s overestimated how often they themselves used informal interpreters and underestimated the patient’s satisfaction with the interpreter quality [23].

According to telephone interviews with general practitioners in Ireland, 77% responded saying language assistance was required during consultation with refugees and asylum seekers [36]. However, the results from the study show that only 7% of HCPs could name a professional interpreting service and only 5% could name one which they had used. In consultations where an interpreter was required but they managed without, the HCP either used sign language and diagrams, the patient spoke some English or the GP themselves had some knowledge of the patient’s language [36]. There was also a greater preference for informal interpreters and the main reason reported was accessibility. However, concerns about confidentiality with informal interpreters was reported by 43% compared to 11% with professional interpreters.

### 3.2. Clinician Cues

Across a number of included studies, patients consistently emphasized the importance of non-verbal cues and compassion from the HCPs, such as smiling, nodding, kindness and showing patience. They were all seen to be factors in helping to alleviate stress and improving trust as they allowed the patients to feel welcome and valued, and reportedly affected perceived levels of engagement [24,26,27,28,29,37,39].


*“When you sit with a doctor and you hear kind words, that has an influence on your nerves, on your body. You start feeling better, healthier, than when the doctor is angry.”*
*—patient* [27]


*“We don’t have anybody here. It is very important that the doctor is friendly.”*
*—patient* [28]

On the other hand, lack of interest from the HCP and not being taken seriously about their health concerns led patients to be less open in their communication [24,26,28,29,42]. Patients reported that they were not likely to trust and communicate with an HCP who was not willing to consider their individual characteristics and needs [26,27,28,42].


*“I did not give him the medical file, because he was not interested. My expectation was somebody who will be open to me, like doctors in Africa.”*
*—patient* [26]


*“That generalizing attitude is what still makes me angry.”*
*—patient* [26]

In contrast, HCP’s willingness to listen to the patient’s personal story and non-medical information was seen as a way to encourage trust and improve the relationship. The HCP’s openness, understanding and attentiveness towards the patient’s needs, alongside willingness to take detailed medical history, helped to build trust and allowed the patient to open up to them [24,26,27,28,29,37].


*“To show that you are interested in the person, not only in the disease; to show that you want to know something about the context. Sometimes it is difficult to find time for it in a busy practice, but I see it is a worthwhile investment…”*
—HCP [27]

### 3.3. Cultural Understanding

Cultural considerations play a key role in open communication and understanding of medical context between patients and HCPs.

When organizing professional interpreters, it was important to some patients that same gender interpreters were organized to allow them to be open with the HCP. When they had interpreters of the opposite gender, they expressed that they felt it was inappropriate and that they felt embarrassed [23,24,28,31,38]. Patients reported that having interpreters and HCPs of the same gender allowed for them to form a connection and speak more freely about their health concerns [24,37].


*“Give her a woman translator, so that she can be open to tell all the problems”*
*—patient* [24]


*“Religion sometimes says it is good for you to have [a] female doctor if you are female”*
*—patient* [24]

HCPs expressed that often a challenge for them was causing patients to understand and explain their symptoms due to cultural differences [23,25,27,30,33,34]. They reported that there were cultural differences in the way some patients interpreted health and illness, as well as challenges in addressing long-standing cultural beliefs which impacted the medical care they gave. Patients also expressed not wanting to contradict the HCPs who were seen as authority figures and felt that any self-advocacy from them would not be accepted, which highlighted the notion of hierarchy within the interaction [31]


*“They have a different culture, so their cultural perception of symptoms and what they mean... trying to interpret the difference between a bloated abdomen and a painful abdomen, just becomes an impossible task...”*
—*HCP* [25]

## 4. Discussion

This review found that refugees and asylum seekers experience a range of communication challenges and obstacles in primary care consultations. These relate to the availability and access to appropriate interpreters, HCP demeanor and cultural considerations. The highlighted themes: linguistic barriers, clinician cues and cultural understanding, are all interrelated and emphasize the preferences for considerate and appropriate care.

While previous research looking at the use of interpreters in healthcare services has shown the benefits of professional interpreters in communication, clinical outcomes, utilization and satisfaction, [43,44] the findings from this review highlight the practical and relational challenges of organizing and using interpreters in consultations with refugees and asylum seekers. Patient preferences for same-sex interpreters further complicated these challenges. Although quantitative studies included in this review indicate the challenge of being able to access professional interpreters, who were more proficient in medical terminology, the qualitative evidence demonstrates that the alternative (i.e., to use informal interpreters) can produce poor quality translation and confidentiality concerns. Importantly, studies included in the review also report concerns about accuracy and confidentiality when using professional interpreters, illustrating that the clinical encounter is complex and that both professional and informal interpreters provide benefits and challenges. Challenges with language and the use of interpreters, for example, transcend clinical context and are a pervasive system challenge [45].

Issues around cultural considerations and understanding were identified as potential challenges in the healthcare encounter. Our review indicates that HCPs often play a role in helping bridge the gap in different cultural understandings but perceive this to be an ongoing challenge in their practice. Other studies in this review focused on cultural issues of gender concordance, and existence of a clinician–patient power dynamic in primary consultations which limited communication. While these cultural issues are undeniably important, previous research highlights that there are many other cultural differences and beliefs which influence health and healing practices [46]. Different cultures have different understandings of illness and disease and many have traditional healing practices [46]. The fact that these issues are absent from the studies included in this review suggest that the research in primary care communication may have only looked at this aspect of communication and the HCP’s role superficially with this population group. To address this research gap, further work should be done to understand the role of cultural factors in developing a shared understanding of health in primary care.

As well as identifying challenges, this review also uniquely summated the literature about factors which facilitate primary care consultations with refugees and asylum seekers. Non-verbal and compassionate care aspects of communication, for example, emerged as an important factor in helping improve comfort and trust between the HCP and patient. The patients preferred to see HCPs who were welcoming, kind and patient, and those who were willing to take time to listen to non-medically relevant information and took an interest in them as a person. These findings align with the previous literature which identifies such non-verbal cues as a method to help alleviate anxiety and improve trust in patient-centered communication. [47,48,49] Non-verbal cues and compassionate care by HCPs play a key role in assisting to build the HCP–patient relationship, and additionally, identify an opportunity for a positive healthcare encounter when there are linguistic and cultural barriers present with patients from refugee and asylum seeker backgrounds.

The refugee and asylum seeker experiences identified in our review are similar to those found in other migrant groups, including language barriers, interaction with HCPs and cultural differences in healthcare [50,51]. Experiences of non-migrant and non-refugee populations also highlight similar desires for the traits which they consider important in their HCP, in terms of clinician demeanor and competence [24,52]. The model of humanistic medicine provides a framework for understanding these similarities as it illustrates that the experiences and preferences of patients are generalizable to the patient experience as a whole. With an emphasis on HCPs being compassionate and empathetic towards their individual patients and being aware of their emotions, concerns and suffering [53], humanistic medicine is seen as the basis of medicine [54]. However, there are challenges with applying humanistic care in practice as HCPs find bureaucratic barriers and challenges with time given the business-like climate of certain areas of medicine [55]. In addition to the linguistic and cultural barriers, HCPs treating refugee and asylum seekers have to navigate social factors and experiences of trauma [5,6]. Nevertheless, applying this framework of humanistic care has benefits to both the patients and the HCPs [55], suggesting organizational support should be given in this area. Greater effort should be undertaken to provide humanistic and compassionate care when encountering refugees and asylum seekers and healthcare systems need to provide support to HCPs to facilitate this approach.

There are strengths and limitations of this review. A strength was that systematic searches were conducted using seven relevant databases with additional reference and citation searches. In addition, full texts which were in languages other than English were also reviewed, further strengthening the search strategy. It is therefore unlikely that published studies have been missed. However, due to the defined inclusion criteria, some literature may have been excluded if it used the broad term of migrants rather than specifying the subpopulation group.

Another strength of this review is that the included literature covers various ethnic groups in various western resettlement countries. However, the number of participants combined from all the studies is still relatively limited which may not allow for any conclusions concerning the communication experiences of a broad group of refugees and asylum seekers in different countries. Furthermore, interpretations based off participant demographics, such as sex difference, age difference or the educational difference in refugee and asylum seeker populations are not possible as they are not reported in many of the included studies. Another limitation is at the search strategy only identified the scientific literature and failed to capture grey literature, such as non-government organization reports which often report on patient and HCP experiences.

## 5. Conclusions

Primary care HCPs need additional support to allocate time and provide compassionate and humanistic care desired by refugees and asylum seekers. Ongoing issues with organizing and routinely utilizing professional interpreters suggest infrastructure should be in place to allow HCPs to be trained on the accessibility of accessing professional interpreters, with systems that allow for timely scheduling. Beyond issues of language, refugees and asylum seekers may also to be sensitized to non-verbal cues and compassionate care from the HCP. This is an area that should be further investigated, particularly in light of the current shift to virtual consultation for some healthcare encounters.

## Figures and Tables

**Figure 1 ijerph-18-01469-f001:**
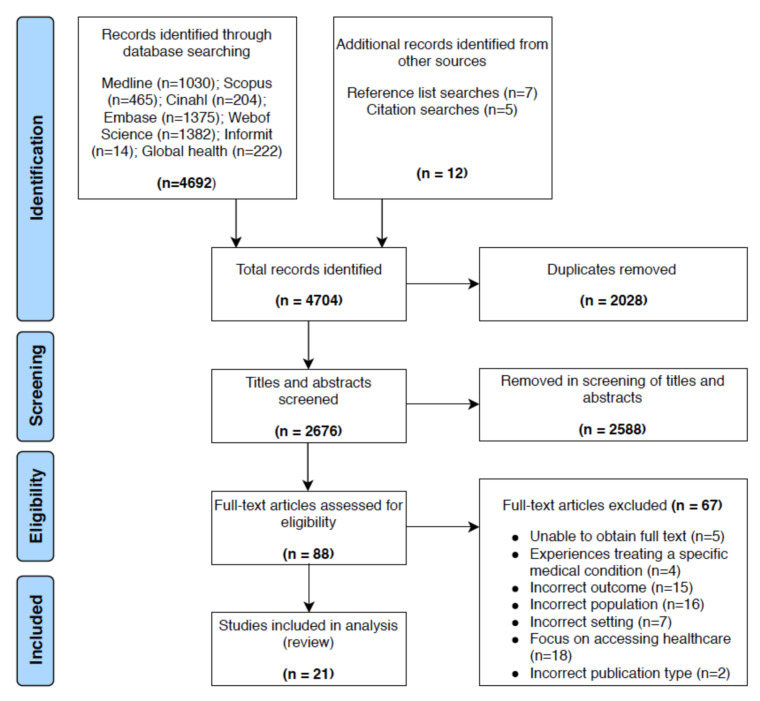
PRISMA flow diagram: The PRISMA diagram details the search strategy and selection process.

**Table 1 ijerph-18-01469-t001:** Summary of participant and study characteristics of included studies.

Author	Year	Country	Population (Service Users)	Setting (Service Provider/Setting)	Number	Data Collection Method	Analysis Methodology	Quality Score
Adair et al. [23]	1999	United States of America	Refugees (Somali)	Primary care clinic—both doctors and nurses	38 patients, 6 nurses, 32 doctors	Refugees—semi-structured telephone interviews Medical professional-survey questions	Quantitative analysis	8
Carroll et al. [24]	2007	United States of America	Refugee women (Somali)	Primary care provider	34 refugees	Refugees—in depth interviews	Grounded theory	9
Farley et al. [25]	2014	Australia	Newly arrived refugees	General Practitioners, nurses, admin staff	20 GPs ^b^, 5 nurse, 11 admin staff	HCP ^c^—focus groups and semi-structured interview	Inductive thematic analysis	9
Feldmann et al. [26]	2006	Netherlands	Refugees (Somali)	General Practitioners	36 refugees	Refugees—in depth interviews	Thematic analysis	7
Feldmann et al. [27]	2007	Netherlands	Refugees (Afghan/Somali)	General Practitioners	66 Refugees, 24 GPs	Refugees—in depth interviews GPs—semi structured interviews	Thematic analysis	7
Feldmann et al. [28]	2007	Netherlands	Refugees (Afghan)	General Practitioners	30 refugees	Refugees—in depth interviews	Thematic analysis	7
Feldmann et al. [29]	2007	Netherlands	Refugees (Afghan/Somali)	General Practitioners	24 GPs	Interviews (refugees and GPs)	General narrative	6
Grut et al. [30]	2006	Norway	Refugees	General Practitioners	12 GPs	GP—interviews	Narrative synthesis	6
Gurnah et al. [31]	2011	United States of America	Refugee women (Somali Bantu)	Reproductive health service	14 refugees	Refugee—interviews, focus group and semi-structured survey	Thematic analysis	8
Harris [6]	2018	Australia	Refugees	General Practice	n/a ^a^	n/a	Opinion article	6
Harris and Zwar [32]	2005	Australia	Refugees	General Practice	n/a	n/a	Opinion article	6
Jensen et al. [33]	2013	Denmark	Refugees	General Practitioners	9 GPs	GP—semi structured interviews	Content analysis	8
Johnson et al. [34]	2008	Australia	Refugees	General Practitioners	12 GPs	GP—semi structured interviews	Template analysis	8
MacFarlane et al. [35]	2009	Ireland	Refugees and asylum seekers	General Practitioners	26 refugees	Refugees—semi-structured interviews	Thematic analysis	9
MacFarlane et al. [36]	2008	Ireland	Refugees and asylum seekers	General Practitioners	56 GPs	GP—telephone survey	Quantitative analysis	8
Manchikanti et al. [37]	2017	Australia	Refugees (Afghan)	General Practice	18 refugees	Refugees—in depth, semi-structured interviews	Thematic analysis	8
Mengesha et al. [38]	2018	Australia	Refugees	General Practitioners, nurses, midwife	5 GPs, 8 nurses, 1 midwife	HCP—semi-structured interviews	Thematic analysis	8
O’Donnell et al. [39]	2008	Scotland (UK)	Asylum seekers	General Practice	52 refugees	Asylum seekers—focus groups and semi-structured interview	Thematic analysis	9
O’Donnell et al. [40]	2007	Scotland (UK)	Asylum seekers	General Practice	52 refugees	Asylum seekers—focus groups, one-on-one interviews or group interviews	Thematic analysis	9
Pottie [41]	2007	Canada	Refugees	Family physician	1 refugee	Refugee—case report		5
Svenberg et al. [42]	2011	Sweden	Refugees (Somali)	General Practice	20 refugees	Refugee—interviews	hermeneutic approach	7

^a^ Abbreviation: n/a, not applicable. ^b^ Abbreviation: GP, general practitioner. ^C^ Abbreviation: HCP, Healthcare provider.

**Table 2 ijerph-18-01469-t002:** Study aims, objectives and outcome measures of included studies.

Author	Study Aims and Objectives	Outcomes Measures	Study Outcomes/Conclusions
Adair et al. [23]	To identify barriers to healthcare access perceived by a group of refugees from Somalia and by the doctors and nurses providing care for them.	Somali and HCP^a^ responses to questions regarding transportation to clinic, payment for medical care, availability of interpreters and satisfaction with the level of communication achieved, comfort with being examined, and obtaining of medical care at multiple clinics.	Nurses and doctors who provide care for these patients and are quite familiar with their demographic characteristics but were inaccurate in predicting how they felt about access to care.
Carroll et al. [24]	To identify characteristics associated with favourable treatment in receipt of preventive healthcare services, from the perspective of resettled African refugee women.	African refugee women’s response to questions about positive and negative experiences with primary healthcare services, beliefs about respectful vs. disrespectful treatment, experiences of racism, prejudice or bias, and ideas about removing access barriers and improving healthcare services.	Qualities associated with a favorable healthcare experience included effective verbal and nonverbal communication, feeling valued and understood, availability of female interpreters and clinicians and sensitivity to privacy for gynecologic concerns.
Farley et al. [25]	To explore the experiences of general practices working within this new model, focusing on the barriers and enablers they continue to experience in providing care to refugees.	HCP responses to questions regarding barriers and enablers experienced when providing refugee healthcare and the resources providers felt would assist them in this task.	HCP working with refugees were enthusiastic and committed. The flexibility of the general practice setting enables providers to be innovative in their approach to caring for refugees. However, most practices continue to feel isolated as they search for solutions.
Feldmann et al. [28]	What are participants’ frames of reference, in respect of healthcare, and what is their definition of health? How did participants try to solve their health-related problems and what was their experience of the process? What personal and social resources were useful to them? How can we explain differences between participants’ experiences of healthcare and their interpretations of their experiences?	Refugee responses to questions regarding healthcare experiences, health-related problems and social and personal resources used in healthcare.	The elements that constituted positive and negative episodes and led to the development or undermining of trust were identified in the narratives. Negative experience tended to be interpreted as a sign of prejudice on the part of the HCP.
Feldmann et al. [26]	Which frames of reference play a role in the development over time of an individual refugee’s relationship with the Dutch healthcare system, in particular with the GP?	Refugee responses to questions regarding healthcare in country of origin and healthcare in the Netherlands.	For a positive relationship to develop, based on trust, GPs need to invest in the relationship with individual refugees, and avoid actions based on prejudice.
Feldmann et al. [29]	What do refugees and general practitioner say about physically inexplicable somatic complaints?	GPs’ perspectives on medically unexplained physical symptoms presented by their refugee patients, strategies to address this and problems assisting refugee patients.	The personal attitude and communication skills of the practitioner appear to be central to building or undermining trust.
Feldmann et al. [27]	To confront the views of refugee patients and general practitioners in the Netherlands, focusing on medically unexplained physical symptoms.	Refugees’ perspectives on health, illness and mental worries, their expectations from doctors and problems dealing with Dutch doctors. GPs’ perspectives on medically unexplained physical symptoms presented by their refugee patients, strategies to address this and problems assisting refugee patients.	GPs need to invest in the relationship with individual refugees, and avoid actions based on prejudice.
Grut et al. [30]	What challenges do the regular GPs experience in meeting these patients (refugee backgrounds)?	GP responses to questions about the challenges about meeting patients from refugee backgrounds.	GPs need more guidance materials to adapt to cultural challenges of treating refugee patients.
Gurnah et al. [31]	Explore the reproductive health experiences of Somali Bantu women in Connecticut, to identify potential barriers to care experienced by marginalized populations.	Somali women’s response to questions regarding perceptions of barriers to reproductive healthcare.	There was a lack of cultural fluency between patients and provider. There is a need for developing cultural competency in health care delivery.
Harris and Zwar [32]	n/a^c^	n/a	Refugees and asylum seekers come to Australia with a range of health problems related to their experience both overseas and in Australia. These problems need to be addressed in general practice, as should preventive care, which is often overlooked.
Harris [6]	n/a	n/a	Need for more integrated health service provision for people from refugee backgrounds, based on trust and communication.
Jensen et al. [33]	To investigate how general practitioners experience providing care to refugees with mental health problems.	GP responses to questions regarding delivery of care to immigrants in general, and delivery of care to patients with different immigration status.	Findings suggest that the development of conversational models for general practitioners including points to be aware of in the treatment of refugee patients may serve as a support in the management of refugee patients in primary care.
Johnson et al. [34]	To document the existence and nature of challenges for GPs who do this work in South Australia. To explore the ways in which these challenges could be reduced. To discuss the policy implications of this in relation to optimising the initial healthcare for refugees	GP responses to questions regarding challenges in providing initial care to refugees, suggestions on how to reduce challenges and ways to optimise initial healthcare for refugees.	GPs in this study were under-resourced, at both an individual GP level as well as a structural level, to provide effective initial care for refugees.
MacFarlane et al. [35]	Exploration of the elements of that experience in terms of their access to informal interpreters, choices and trade-offs about who to ask and negotiations with general practitioners about their use.	Asylum seeker responses to questions around use of health services; barriers and facilitators to accessing care; use of secondary care services; experience of translators; and previous experience of healthcare in responders’ country of origin.	Overall, service users experience a tension between the value of having someone present to act as their interpreter and the burden of work and responsibility to manage the language barrier.
MacFarlane et al. [36]	Quantify the need for language assistance in general practice consultations and examine the experience of, and satisfaction with, methods of language assistance utilised.	GP^b^ responses to questions regarding the need for language assistance, their knowledge and use of professional interpreters and use of informal interpreters	The need for language assistance in consultations with refugees and asylum seekers in Irish general practice is high. General practitioners rely on informal responses.
Manchikanti et al. [37]	To investigate the acceptability of general practitioner (GP) services and understand what aspects of acceptability are relevant for Afghan refugees.	Refugees responses to questions regarding access to primary healthcare.	The findings reinforce the importance of tailoring healthcare delivery to the evolving needs and healthcare expectations of newly arrived and established refugees, respectively.
Mengesha et al. [38]	To explore the healthcare professional (HCP) experiences of working with interpreters when consulting refugee and migrant women who are not proficient in English around sexual and reproductive health issues.	HCP responses to questions regarding their recent encounters with refugee and migrant women not proficient in English language in sexual and reproductive healthcare.	Communication barriers in the provision of sexual reproductive health services to refugee and migrant women may not be avoided despite the use of interpreters.
O’Donnell et al. [39]	How migrants’ previous knowledge and experience of healthcare influences their current expectations of healthcare in a system relying on clinical generalists performing a gatekeeping role.	Asylum seekers response to health services; barriers and facilitators to accessing care; use of secondary care services; experience of translators; and previous experience of health care in responders’ country of origin.	HCPs need to be aware that experience of different systems of care can have an impact on individuals’ expectations in a GP- led system.
O’Donnell et al. [40]	To identify the barriers and facilitators to accessing healthcare, both medical and dental, and to explore the healthcare needs and beliefs of asylum seekers.	Asylum seeker responses to discussion around health services; barriers and facilitators to accessing care; use of secondary care services; use of dental services; experience of translators; and previous experience of healthcare in their own country.	The findings highlight issues of access to timely health care and the role of interpreters within the consultation. In addition to understanding the role of GPs and the UK health system.
Pottie [41]	n/a	n/a	The quality of patient care is improved with the use of professional interpreters.
Svenberg et al. [42]	To explore Somali refugees’ experience of their encounters with Swedish healthcare.	Refugees’ responses to questions regarding their and their family’s experience with meeting Swedish healthcare.	Interpretation of the findings suggests unfulfilled expectations of the medical encounters, resulting in disappointment among the Somali informants. This entailed a lack of trust and feelings of rejection and, ultimately, decisions to seek private medical care abroad.

^a^ Abbreviation: HCP, Healthcare provider. ^b^ Abbreviation: GP, general practitioner. ^C^ Abbreviation: n/a, not applicable.

## Data Availability

Not applicable.

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
