# Peer review of "(untitled)"

_ijerph, 2021, doi:10.3390/ijerph18041469_

Round 1
Reviewer 1 Report
The manuscript titled "Communication experiences in primary healthcare with refugees and asylum seekers: A literature review and narrative synthesis" is an attempt to consolidate the available data on the challenges encountered during the interactions between caregivers and refugees or asylum seekers. The study emphasizes on the sensitization of the healthcare providers while dealing with these patients facing challenging circumstances and from a different cultural background. It also highlights the role of interpreters in bridging this gap and provide a better quality of healthcare to refugees.
Strengths:
- Comprehensive literature analysis.
- Well laid out methodology.
- Well defined screening procedure.
Weakness:
- Not much emphasis on interpreters' (both formal and informal) point of view.
Author Response
Reviewer 1:The manuscript titled "Communication experiences in primary healthcare with refugees and asylum seekers: A literature review and narrative synthesis" is an attempt to consolidate the available data on the challenges encountered during the interactions between caregivers and refugees or asylum seekers. The study emphasizes on the sensitization of the healthcare providers while dealing with these patients facing challenging circumstances and from a different cultural background. It also highlights the role of interpreters in bridging this gap and provide a better quality of healthcare to refugees.
Strengths:
1.Comprehensive literature analysis.
2.Well laid out methodology.
3.Well defined screening procedure.
Weakness:1.Not much emphasis on interpreters' (both formal and informal) point of view.
Response: Thank you for your positive feedback. Our search strategy was purposefully focused on finding health care provider and patient perspectives on communication, and it did not capture literature which reported outcomes from the interpreters’ point of view. The inclusion of the interpreter’s point of view would be an additional research question and was not within the scope of this review.
Reviewer 2 Report
Dear researchers,
Thanks a lot for giving me the possibility to review your paper "Communication experiences in primary healthcare 1 with refugees and asylum seekers: A literature review 2 and narrative synthesis" in this Pandemic Period.
I read your paper and I find it very interesting, but however, it need of more improvements :
it's a literature review but I find it not complete. I have expected a better organization in the analysis that need to be improved. I would have expected a better care in the format (the final table has to be reorganized, it can't take all these pages). Why did'nt you develop a bibliometric complete analysis?
I found also the conclusions quite weak.
Author Response
Thanks a lot for giving me the possibility to review your paper "Communication experiences in primary healthcare 1 with refugees and asylum seekers: A literature review 2 and narrative synthesis" in this Pandemic Period.
I read your paper and I find it very interesting, but however, it needof more improvements:it's a literature review but I find it not complete. I have expected a better organization in the analysis that need to be improved.I would have expected a better care in the format (the final table has to be reorganized, it can't take all these pages). Why didn’t you develop a bibliometric complete analysis?
Response: Thank you for your suggestions. For the review we have used a narrative synthesis methodology as described by Popay et al.This methodology was purposefully selected as quantitative, qualitative and opinion papers were all included in our review. The thematic analysis involved generating codes from the literature to identify key ideas and then identifying the themes by grouping the codes with similar ideas together. The relevant codes which aligned with the initial research question were all incorporated into themes. Bibliometric complete analysis was not conducted in this review as a narrative synthesis methodology allowed the qualitative analysis of the included studies.
In line with your suggestion, we have now amended Table 1in the manuscript. Specifically, we have replaced it with two separate tables. Table 1 now shows the study and participant characteristics and Table 2 now shows the study aims, objectives and outcome measures for the studies included in our review. We also edited the tables so that they do not use double spacing and are organized alphabetically by author.
I found also the conclusions quite weak.
Response: We have now amended our conclusions, in lines 404onwards in the manuscript, to highlight the importance of non-verbal cues and compassionate care from the HCP.
Reviewer 3 Report
Dear authors
This is a very good topic to elucidate the challenges the refugees and asylum seekers have faced. The paper is very straight-forward. Here are some suggestions for you and please take them into consideration for further possibility of publication. English should be well revised.
52 this is a good paragraph to illustrate the healthcare service, however, you end up with “communication is likely to play a key role in the healthcare encounter….” But, I think you can address more to “explain” why the communication is so important as a key to figure out something.
94 all things are good, however “undocumented migrants” should be carefully defined in more detail to make the readers clear.
Table 1 I know “GP” means general practitioner, but still needs the description of abbreviation
103/104 these are two very good examples, but I think you can mention a little bit more reference here to emphasize this point, “gender matters”.
140 “this suggests that perhaps the research in primary” è I can not catch up with this meaning in the sentence, please clarify the meaning more clearly.
160 “experiences of non-migrant and non-refugee populations” è not sure why the population mentioned here, do you wan to highlight the difference, such as the clinician demeanour and competence?
182 regarding the limitation in the literature, could you please provide the sex difference, age difference or the educational difference in the refugee and asylum seeker populations to face the problems, if not, that could be one of the limitations.
Author Response
This is a very good topic to elucidate the challenges the refugees and asylum seekers have faced. The paper is very straight-forward. Here are some suggestions for you and please take them into consideration for further possibility of publication. English should be well revised.
Response: Thank you for your comment. The manuscript has been proofread by multiple authors to ensure that the English language and style are consistent throughout the manuscript.
52 this is a good paragraph to illustrate the healthcare service, however, you end up with “communication is likely to play a key role in the healthcare encounter....” But, I think you can address more to “explain” why the communication is so important as a key to figure out something.
Response: Thank you for your feedback. We have now amended the sentence in the manuscript to elaborate this point further. “Communication is likely to play a key role in the healthcare encounter between refugee and asylum seekers and healthcare providers, as communication is an essential starting point for patient satisfaction and positive health outcomes.”
94 all things are good, however “undocumented migrants” should be carefully defined in more detail to make the readers clear.
Response: Thank you for your feedback. We have now amended the sentence to include a definition of undocumented migrants. “ ‘Undocumented migrants,’ defined as anyone residing in any given country without legal documentation, were also excluded...”
Table 1I know “GP” means general practitioner, but still needs the description of abbreviation
Response: Thank you for your suggestion. We have now added a footnote to table 1, describing the abbreviations within the table.
103/104 these are two very good examples, but I think you can mention a little bit more reference here to emphasize this point, “gender matters”.
Response: Thank you for your comment. We agree that gender is an important construct, however, this did not feature in the data beyond gender concordance for interpreters and healthcare provider’s. We expect that this is due to the area being under reportedin literature looking at communication between healthcare providers and refugee and asylum seekers, and have noted as such in our discussion, in lines 160-170.
140 “this suggests that perhaps the research in primary” I can not catch up with this meaning in the sentence, please clarify the meaning more clearly.
Response: We have now amended the paragraph, in lines 164-170ofthe manuscript
160 “experiences of non-migrant and non-refugee populations” not sure why the population mentioned here, do you wantto highlight the difference, such as the clinician demeanour and competence?
Response: Thank you for your comment. We want to highlight that clinician demeanour and competence is a universal desire by patients regardless of their background. Positive clinician cues and demeanour are well regarded by refugee and asylum seeker patients but also by patients of non-migrant and non-refugee population groups.
182 regarding the limitation in the literature, could you please provide the sex difference, age difference or the educational difference in the refugee and asylum seeker populations to face the problems, if not, that could be one of the limitations.
Response: Thank you for your suggestion. As this review reports on both qualitative and quantitative papers, the participant demographics reported in the included papers are provided as ranges. It would not be appropriate or possible to report of these factors from the qualitative datasets. We have now added this as one of the limitations, in lines 403-405 of the manuscript.
Reviewer 4 Report
The paper is well planned and written, the target population is precisely and correctly chosen, the literature search is very well conceived and conducted.
The logic is clear and the sections are concise and at the same time complete and well understandable.
The only weak point I see in this version of the paper is the Table 1, which is full of interesting data but it is presented in a format which doesn’t at all help the reader to browse and understand them.
Apart from the double spacing, which forces the table to span in 11 pages (definitely too many…), the order in which the articles are presented (alphabetic order of the main author) it is not wrong “per se”, but it asks the reader to go back and forth in this very long table to try to have an idea of the time evolution of the studies, or to find studies with a common methodology, or countries, etc..
In my opinion, if the authors choose a less basic order logic for this table (eg. (a) time span -> from the oldest to the most recent year of publication; or (b) Analysis methodology, followed by time span; or (c) Quality order -> from highest quality to lowest ranked quality, etc. – and possible various combinations of those examples), the comparisons and the understanding of the table will improve a lot, becoming a better companion of an otherwise very enjoyable paper.
Another option may be to split the table in two different tables, ordered both by year of publications but, for example, one focused on Methodology and Quality scores and the other one on the other data, just to avoid to “overload” the reader and give an immediate overview about which study design produced higher quality data (why not using green-yellow-orange colors for the quality scores to help an eyeball perception ?) and help readers to focus, in the second table, on more in-depth information about the studies which attracted their interest in the first one…
Finally, in the first row of the abstract, it seems that a verb “are” lacks somewhere. I will understand better the first sentence of the abstract in the form “Refugee and asylum seeker populations ARE rising in Western Countries”.
Author Response
The paper is well planned and written, the target population is precisely and correctly chosen, the literature search is very well conceived and conducted.The logic is clear,and the sections are concise and at the same time complete and well understandable.
Response: Thank you for your comment.
The only weak point I see in this version of the paper is the Table 1, which is full of interesting data, but it is presented in a format which doesn’t at all help the reader to browse and understand them.
Apart from the double spacing, which forces the table to span in 11 pages (definitely too many...), the order in which the articles are presented (alphabetic order of the main author) it is not wrong “per se”, but it asks the reader to go back and forth in this very long table to try to have an idea of the time evolution of the studies, orto find studies with a common methodology, or countries, etc.
In my opinion, if the authors choose a less basic order logic for this table (eg. (a) time span -> from the oldest to the most recent year of publication; or (b) Analysis methodology, followedby time span; or(c) Quality order -> from highest quality to lowest ranked quality, etc. –and possible various combinations of those examples), the comparisons and the understanding of the table will improve a lot, becoming a better companion of an otherwise very enjoyable paper.
Another option may be to split the table in two different tables, ordered both by year of publications but, for example, one focused on Methodology and Quality scores and the other one on the other data, just to avoid to “overload” the reader and give an immediate overview about which study design produced higher quality data (why not using green-yellow-orange colors for the quality scores to help an eyeball perception ?) and help readers to focus, in the second table, on more in-depth information about the studies which attracted their interest in the first one...
Response: Thank you for your suggestions. We have now split the table into two separate tables. Table 1now shows the study aims, objectives and outcomes measures and table 2 now shows the study and participant characteristics. We also edited the tables so that they do not use double spacing and are organised alphabetically by author.
Finally, in the first row of the abstract, it seems that a verb “are” lacks somewhere. I will understand better the first sentence of the abstract in the form “Refugee and asylum seeker populations ARE rising in Western Countries”.
Response: Thank you for bringing this to our attention. We have edited the sentence in the manuscript to now read “Refugee and asylum seeker populations are rising in Western Countries.”
Round 2
Reviewer 2 Report
Congratulations
Author Response
Thank you for your comment